# Visual experience shapes functional connectivity between occipital and non-visual networks

**Mengyu Tian[1,2]\*, Xiang Xiao[3], Huiqing Hu[4], Rhodri Cusack[4], Marina Bedny[2]**

[1]Center for Educational Science and Technology, Beijing Normal University, Zhuhai, China; [2]Department of Psychological and Brain Sciences, Johns Hopkins University, Baltimore, United States; [3]Department of Psychology, Faculty of Art and Science, Beijing Normal University at Zhuhai, Zhuhai, China; [4]Trinity College Institute of Neuroscience and School of Psychology, Trinity College Dublin, Dublin, Ireland

**\*For correspondence:**
mengyutian@jhu.edu

**Competing interest:** The authors declare that no competing interests exist.

## eLife Assessment

This **important** study provides evidence supporting the hypothesis that postnatal visual experience shapes the patterns of functional connectivity between extrastriate visual cortex and frontal regions, by comparing neonates, blind and sighted adults using resting-state fMRI. The evidence supporting the main claim is **convincing**, and the authors' interpretations are appropriately calibrated in the discussion. Nevertheless, the study design and methodology are inherently limited to resolve the underlying mechanisms driving connectivity changes during neurodevelopment (experience-related plasticity vs post-natal experience-independent maturation). This study will be of broad interest to neuroscientists and neuroimaging researchers studying vision, plasticity and brain development.

**Abstract** Comparisons of visual cortex function across blind and sighted adults reveal effects of experience on human brain function. Since almost all research has been done with adults, little is known about the developmental origins of plasticity. We compared resting-state functional connectivity of visual cortices of blind adults (*n* = 30), blindfolded sighted adults (*n* = 50) to a large cohort of infants (Developing Human Connectome Project, *n* = 475). Visual cortices of sighted adults show stronger coupling with non-visual sensory-motor networks (auditory, somatosensory/motor) than with higher-cognitive prefrontal cortices (PFC). In contrast, visual cortices of blind adults show stronger coupling with higher-cognitive PFC than with non-visual sensory-motor networks. Are infant visual cortices functionally like those of sighted adults, with blindness leading to functional change? We find that, on the contrary, secondary visual cortices of infants are functionally more like those of blind adults: stronger coupling with PFC than with non-visual sensory-motor networks, suggesting that visual experience modifies elements of the sighted adult long-range functional connectivity profile. Infant primary visual cortices are in between blind and sighted adults, that is, more balanced PFC and sensory-motor connectivity than either adult group. The lateralization of occipital-to-frontal connectivity in infants resembles the sighted adults, consistent with the idea that blindness leads to functional change. These results suggest that both vision and blindness modify functional connectivity through experience-driven (i.e., activity-dependent) plasticity.

## Introduction

Relative to sighted adults, visual cortices of adults born blind show enhanced responses during non-visual tasks, such as reading braille and localizing sounds as well as distinctive patterns of long-range

functional connectivity with non-visual networks (*Abboud and Cohen, 2019*; *Bedny et al., 2011*; *Burton et al., 2012*; *Burton et al., 2014*; *Butt et al., 2013*; *Collignon et al., 2011*; *Deen et al., 2015*; *Kanjlia et al., 2016*; *Kanjlia et al., 2021*; *Lane et al., 2015*; *Liu et al., 2007*; *Sadato et al., 1996*; *Striem-Amit et al., 2015*; *Watkins et al., 2012*). Since almost all research thus far has been done with adults, a key outstanding question concerns the developmental origins of these experience-based differences.

One possibility is that at birth, infant visual cortices start out in the 'prepared' sighted adult state and blindness modifies this functional connectivity. Alternatively, infant visual cortices may start out functionally similar to those of blind adults, and lifetime visual experience shapes connectivity toward the sighted adult pattern. To distinguish between these possibilities, we compare the long-range function connectivity of visual cortices across blind adults, sighted adults, and a large cohort of 2-week-old infants (Developing Human Connectome Project, dHCP, *n* = 475). Using resting-state data provide a common measure of cortical function across these diverse populations.

To our knowledge, no prior studies have compared infants to multiple populations of adults with different sensory experiences. Previous studies comparing infants to sighted adults have largely reported similarity across groups (*Barttfeld et al., 2018*; *Doria et al., 2010*; *Fransson et al., 2009*; *Gao et al., 2009*; *Liu et al., 2008*; *Zhang et al., 2019*). However, these studies focused on whether large-scale functional networks are present in infancy, for example, stronger connectivity of regions within the visual network than between visual and auditory regions. Studies comparing blind and sighted adults find differences across groups in which non-visual networks are most strongly coupled with the visual system, that is, visual cortices of sighted adults show stronger coupling with non-visual sensory-motor networks (i.e., auditory, somatosensory/motor) than higher-cognitive systems; by contrast, in blind adults, visual cortex coupling is stronger with higher-cognitive prefrontal cortices (PFC) than with non-visual sensory-motor networks (*Abboud and Cohen, 2019*; *Bedny et al., 2011*; *Burton et al., 2014*; *Deen et al., 2015*; *Kanjlia et al., 2021*; *Liu et al., 2007*; *Qin et al., 2013*; *Striem-Amit et al., 2015*; *Watkins et al., 2012*; *Yu et al., 2008*). In the current study, we compare this experience-sensitive functional signature across infants, sighted, and blind adults.

We measured the connectivity profile of four occipital 'visual' areas that show cross-modal plasticity in blindness, that is, are active during non-visual tasks in blind people and show related changes in resting-state functional connectivity.

We focused on three functionally distinct secondary visual areas (located in lateral, dorsal, and parts of the ventral occipital cortex) and the primary visual cortex (V1). The three secondary visual areas have been found to respond to different non-visual tasks in blind people: language tasks, numerical reasoning tasks, and executive control tasks, respectively. Enhanced coupling with PFC is observed across all three occipital regions in blind adults. However, each region shows preferential coupling with a distinct subregion of PFC with analogous functional profiles, that is, language responsive occipital areas are more coupled with language responsive PFC (*Kanjlia et al., 2016*; *Kanjlia et al., 2021*; *Lane et al., 2015*). These observations suggest that resting-state and task-based functional profiles are related, although the functional and developmental nature of this relationship remains an open question.

The precise visual functions of the studied secondary visual regions in sighted people are not known. Anatomically, these regions in sighted people approximately correspond to the locations of motion-sensitive V5/MT+ and the lateral occipital complex (LO), as well as ventral portions of occipito-temporal cortex including V4v and dorsal portions including V3a. The occipital region of interest (ROI) also extends ventrally into the middle portion of the ventral temporal lobe and dorsally into the intra-parietal sulcus and superior parietal lobule (*Tootell et al., 1997*; *Van Essen et al., 2001*).

We also examined connectivity of anatomically defined primary visual cortex (V1), which likewise shows altered task-based responses and functional connectivity in congenitally blind adults (*Amedi et al., 2003*; *Bedny et al., 2011*; *Burton et al., 2014*; *Butt et al., 2013*; *Lane et al., 2015*; *Raz et al., 2005*; *Sadato et al., 1996*; *Striem-Amit et al., 2015*; *Yu et al., 2008*). Since many previous studies have found that blindness alters the balance of connectivity between visual cortex and higher-order prefrontal as opposed to sensory-motor regions, this was our primary outcome measure (*Abboud and Cohen, 2019*; *Bedny et al., 2011*; *Burton et al., 2014*; *Deen et al., 2015*; *Heine et al., 2015*; *Kanjlia et al., 2016*; *Kanjlia et al., 2021*; *Lane et al., 2015*; *Liu et al., 2007*; *Sen et al., 2022*; *Striem-Amit et al., 2015*). We also examined changes in connectivity lateralization—that is, the balance of

between versus within hemisphere connectivity, since prior task-based studies have found laterality changes in blindness as well as co-lateralization of occipital and non-occipital networks in this population (*Kanjlia et al., 2021*; *Lane et al., 2017*; *Tian et al., 2023*).

To preview the results, we find that, in infants, the long-range functional connectivity profile of secondary visual areas resembles that of blind adults, whereas V1 falls between blind and sighted adult populations. Relative to sighted adults, both blind adults and infants show stronger coupling between visual cortices and PFC and weaker coupling between visual cortices and non-visual sensory-motor networks. This suggests that vision contributes to modifying the balance of connectivity between occipital and non-visual networks after birth. In contrast, connectivity lateralization patterns appear to reflect blindness-related modification.

## Results

### Connectivity profile of secondary visual cortices in infants is more similar to that of blind than sighted adults

We first examined the long-range functional connectivity of (three) secondary visual areas with sensory-motor areas on the one hand, and higher-order PFC networks on the other. In sighted adults, all three secondary visual areas showed stronger functional connectivity with non-visual sensory-motor areas (primary somatosensory and motor cortex, S1/M1, and primary auditory cortex, A1) than with higher-cognitive PFC. By contrast, in blind adults, all secondary visual regions showed stronger functional connectivity with PFC than with non-visual sensory-motor areas (S1/M1 and A1) (group (sighted adults, blind adults) by ROI (PFC, non-visual sensory) interaction effect: $F_{(1, 78)} = 148.819$, $p < 0.001$; post hoc Bonferroni-corrected paired *t*-test, sighted adults: non-visual sensory > PFC: $t_{(49)} = 9.722$, $p < 0.001$; blind adults: non-visual sensory < PFC: $t_{(29)} = 8.852$, $p < 0.001$; *Figure 1*).

Like in blind adults, in infants, secondary visual areas showed higher connectivity to PFC than to non-visual sensory-motor areas (S1/M1 and A1) (non-visual sensory < PFC paired *t*-test, $t_{(474)} = 20.144$, $p < 0.001$) (*Figure 1*). The connectivity matrix of infants was more correlated with that of blind than sighted adults, but strongly correlated with both adult groups (secondary visual, PFC, and non-visual sensory areas: infants correlated to blind adults: $r = 0.721$, $p < 0.001$; to sighted adults: $r = 0.524$, $p < 0.001$; difference between correlations of infants to blind versus to sighted adults: $z = 3.77$, $p < 0.001$; see *Figure 1—figure supplement 1* for the connectivity matrices).

These results suggest that vision is required to set up elements of the sighted adult functional connectivity pattern, that is, vision enhances occipital cortex connectivity to non-visual sensory-motor networks and dampens connectivity to higher-cognitive prefrontal networks.

We checked the robustness of these results in a number of ways. We first compared the effects across the three secondary visual regions and observed the same pattern across all (*Figure 1—figure supplement 2*). Next, to check the robustness of the findings in infants, we randomly split the infant dataset into two halves and did split-half cross-validation. Across all comparisons, the results of the two halves were highly similar, suggesting the effects are robust (see *Figure 1—figure supplements 3 and 4*). We performed this validation procedure for all analyses reported below with similar results.

### The connectivity pattern of V1 is influenced by early visual experience and blindness

V1 showed the same dissociation between sighted and blind adults as secondary visual areas: in sighted adults, V1 has stronger functional connectivity with non-visual sensory-motor areas than with PFC. By contrast, in blind adults, V1 shows stronger connectivity with PFC than with non-visual sensory areas (group (sighted adults, blind adults) by ROI (PFC, non-visual sensory) interaction: $F_{(1, 78)} = 125.775$, $p < 0.001$; post hoc Bonferroni-corrected paired *t*-test, sighted adults non-visual sensory > PFC: $t_{(49)} = 9.404$, $p < 0.001$; blind adults non-visual sensory < PFC: $t_{(29)} = 7.128$, $p < 0.001$; *Figure 2*).

The pattern for infants in V1 fell between that of sighted and blind adults. The connectivity matrix of infants (V1, PFC, and non-visual sensory) was equally correlated with blind and sighted adults (infants correlated to blind adults: $r = 0.654$, $p < 0.001$; to sighted adults: $r = 0.594$, $p < 0.001$; correlation of infants with blind versus with sighted adults: $z = 0.832$, $p = 0.406$; see *Figure 1—figure supplement 1* for the connectivity matrices). The difference in connectivity strength between V1 to PFC and V1 to non-visual sensory regions was weaker in infants than in sighted or blind adults (group

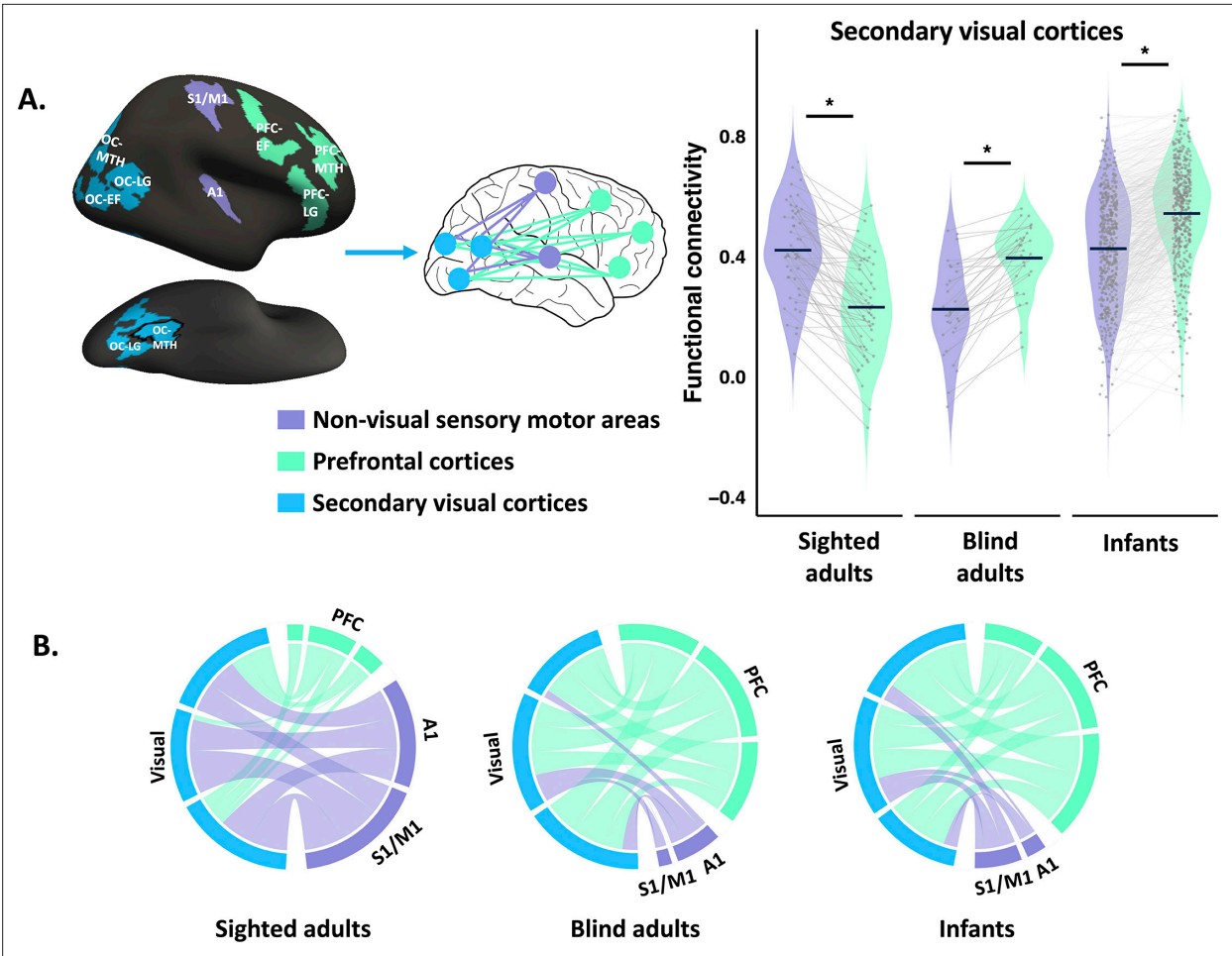

**Figure 1.** Functional connectivity of secondary visual cortices. (**A**) Violin plots show the distributions of functional connectivity (*r*) of secondary visual cortices (blue) to non-visual+++ sensory-motor areas (purple) and prefrontal cortices (green), averaged across three occipital, PFC, and sensory-motor regions of interest (ROIs; A1 and S1/M1) in sighted adults (n = 50), blind adults (n = 30), and infants (n = 475). Individual dots denote mean connectivity values per participant. Gray trend lines illustrate within-participant changes across sensory-motor and prefrontal targets. Dark-blue horizontal markers indicate group averages. ROIs displayed on the left. Note that regions extend to ventral surface, not shown. See ***Figure 1—figure supplement 5*** for the full views of three occipital ROIs. (**B**) Circle plots represent the connectivity of secondary visual cortices to non-visual networks, min–max normalized to [0,1], that is, as a proportion. OC: occipital cortices; MTH: math-responsive region; LG: language-responsive region; EF: executive function-responsive (response-conflict) region. Asterisks (*) denote significant Bonferroni-corrected pairwise comparisons (p < 0.05, see Results section for statistical details). Error bars represent SEM.

The online version of this article includes the following figure supplement(s) for figure 1:

**Figure supplement 1.** The resting-state functional connectivity matrices.

**Figure supplement 2.** Functional connectivity of three secondary visual regions.

**Figure supplement 3.** Infants split-half results.

**Figure supplement 4.** Infants split-half results.

**Figure supplement 5.** Full views of the occipital regions of interest (ROIs).

**Figure supplement 6.** Age-matched adults subgroup results.

**Figure supplement 7.** Results of the dataset excluding infants with radiology scores of 4 or 5.

**Figure supplement 8.** Examples of region of interest (ROI) alignment on individual functional images.

**Figure supplement 9.** Results of datasets excluding adults with signal dropout.

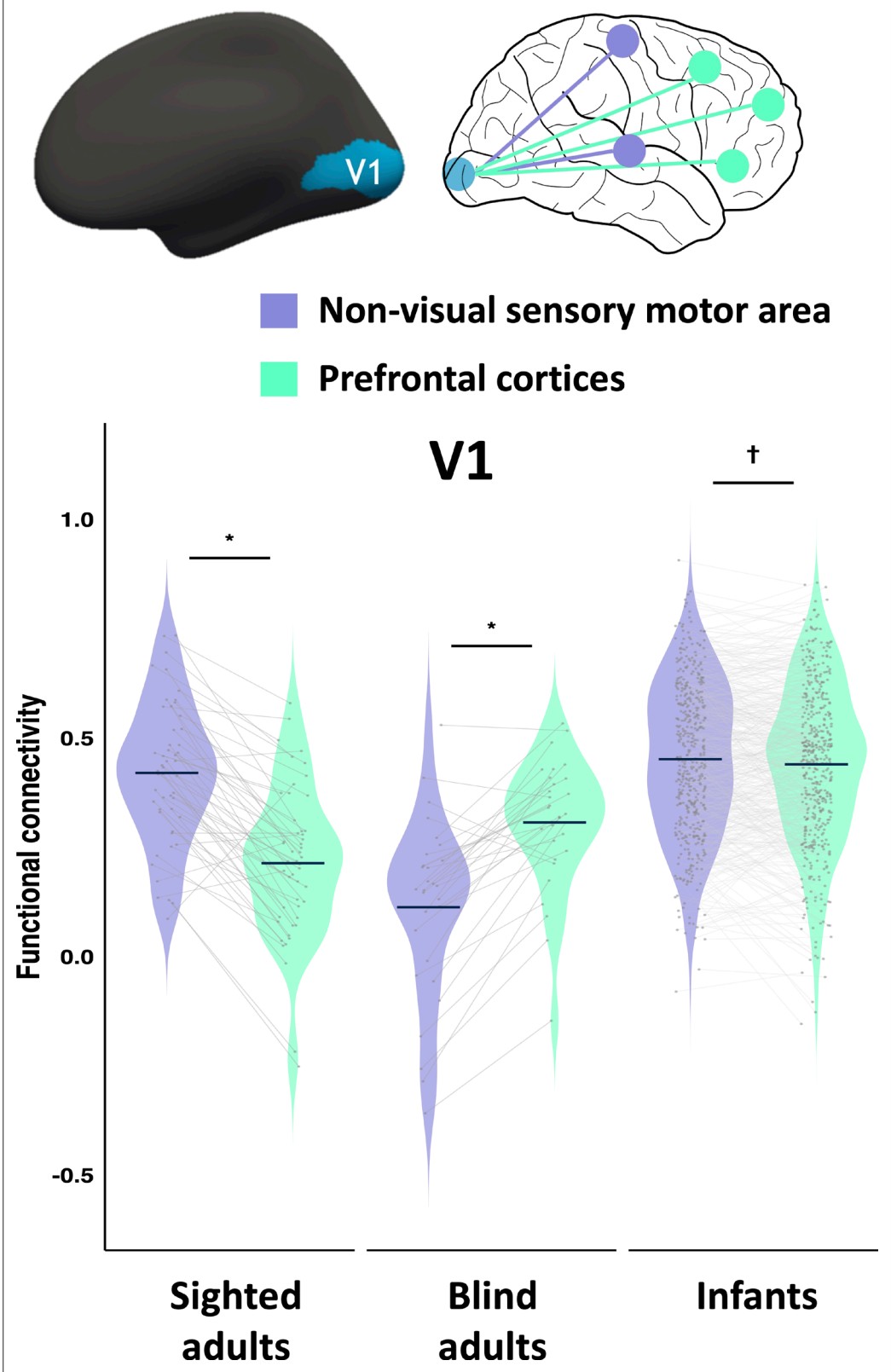

**Figure 2.** Functional connectivity of primary visual cortices (V1). Violin plots show the distributions of functional connectivity (*r*) of V1 to non-visual sensory-motor areas (purple) and prefrontal cortices (green), averaged across three PFC regions of interest (ROIs) and sensory-motor ROIs (S1/M1 and A1) in sighted adults (n = 50), blind adults (n = 30), and infants (n = 475). Individual dots denote mean connectivity values per participant. Gray trend lines

*Figure 2 continued on next page*

*Figure 2 continued*

illustrate within-participant changes across sensory-motor and prefrontal targets. Dark-blue horizontal markers indicate group averages. Asterisks (*) denote significant Bonferroni-corrected pairwise comparisons (p < 0.05). Cross (†) denotes marginal difference in Bonferroni-corrected pairwise comparisons (0.05 < p < 0.1, see Results section for statistical details). Error bars represent SEM.

The online version of this article includes the following figure supplement(s) for figure 2:

**Figure supplement 1.** The correlation of discrepancy in connectivity of visual cortex with age.

**Figure supplement 2.** Preterm and term infant results.

(sighted adults, infants) by ROI (PFC, non-visual sensory) interaction effect: $F_{(1, 523)}$ = 92.21, p < 0.001; group (blind adults, infants) by ROI (PFC, non-visual sensory) interaction effect: $F_{(1, 503)}$ = 57.444, p < 0.001). V1 of infants showed marginally stronger connectivity to non-visual sensory regions (S1/M1 and A1) than PFC (non-visual sensory regions > PFC, paired *t*-test, $t_{(474)}$ = 1.95, p = 0.052; *Figure 2*).

The dHCP cohort included both full-term neonates and preterm infants, scanned at their equivalent gestational age. Visual exposure, therefore, varied somewhat in duration across infants (from 0 to 19.71 weeks), with slightly longer exposure in preterm babies. This variation did not affect connectivity patterns either in V1 or secondary visual cortices (V1: *r* = 0.06, p = 0.192; secondary visual: *r* = 0.004, p = 0.923; see *Figure 2—figure supplement 1*). We also compared the connectivity patterns of preterm (*n* = 90) and full-term infants (*n* = 385) and found no difference from each other or from the all-infant dataset (see *Figure 2—figure supplement 2*). A few weeks of vision after birth is therefore insufficient to influence connectivity.

## Evidence for blindness-related functional change in laterality of occipito-frontal connectivity

Compared to sighted adults, blind adults exhibit a stronger dominance of within-hemisphere connectivity over between-hemisphere connectivity. That is, in people born blind, left visual networks are more strongly connected to left PFC, whereas right visual networks are more strongly connected to right PFC. By contrast, in sighted adults, this lateralized pattern is weaker: visual areas in each hemisphere show only a modest preference for ipsilateral prefrontal cortices, and connectivity with the contralateral PFC remains comparatively strong. This difference between adult groups is observed for both V1 and secondary visual cortices (group (blind adults, sighted adults) by lateralization (within hemisphere, between hemisphere) interaction in secondary visual cortices: $F_{(1, 78)}$ = 131.51, P<0.001; V1: $F_{(1, 78)}$=87.211, P<0.001). Secondary visual cortices showed a significant within >between difference in both groups, with a larger effect in the blind group (post hoc tests, Bonferroni-corrected paired:

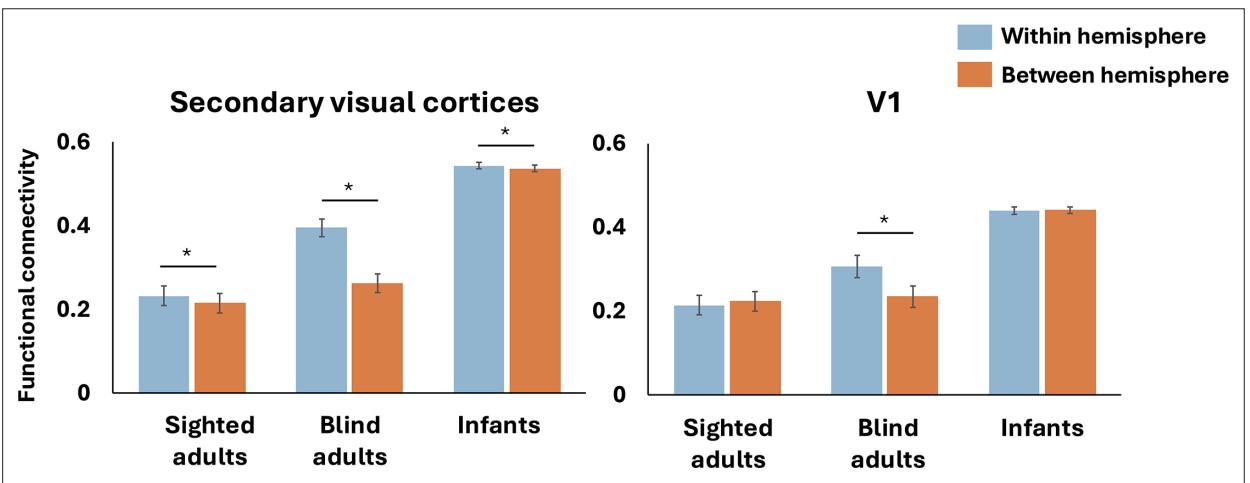

**Figure 3.** Within hemisphere versus between hemisphere functional connectivity. A bar graph shows within hemisphere (blue) and between hemisphere (orange) functional connectivity (*r* coefficient of resting-state correlations) of secondary visual (left) and V1 (right) to prefrontal cortices in sighted adults (n = 50), blind adults (n = 30), and infants (n = 475). Blind adults show a larger difference than any of the other groups. Asterisks (*) denote significant Bonferroni-corrected pairwise comparisons (p < 0.05, see Results section for statistical details). Error bars represent SEM.

*t*-test: sighted adults within hemisphere > between hemisphere: $t_{(49)}$ = 7.441, p = 0.012, Cohen'd = 0.817; blind adults within hemisphere > between hemisphere: $t_{(29)}$ = 10.735, p < 0.001, Cohen'd = 1.96). In V1, only the blind group showed a significant within > between hemisphere effect (post hoc Bonferroni-corrected paired: *t*-test: sighted adults within hemisphere < between hemisphere: $t_{(49)}$ = 3.251, p = 0.101; blind adults within hemisphere > between hemisphere: $t_{(29)}$ = 7.019, p < 0.001).

With respect to laterality, infants resembled sighted more than blind adults (**Figure 3**). For secondary visual cortices, there was a significant difference between blind adults and infants and no difference between sighted adults and infants (group (blind adults, infants) by lateralization (within hemisphere, between hemisphere) interaction effect: $F_{(1, 503)}$ = 303.04, p < 0.001; group (sighted adults, infants) by lateralization (within hemisphere, between hemisphere) interaction effect: $F_{(1, 523)}$ = 2.244, p = 0.135). A similar group by laterality interaction was observed for V1 (group (blind adults, infants) by lateralization (within hemisphere, between hemisphere) interaction: $F_{(1, 503)}$ = 123.608, p < 0.001; group (sighted adults, infants) by lateralization (within hemisphere, between hemisphere) interaction effect: $F_{(1, 523)}$ = 2.827, p = 0.093). This suggests that the enhancement of within over between hemisphere long-range connectivity is related to blindness-driven functional change.

Task-based functional MRI (fMRI) studies find that cross-modal responses in occipital cortex co-lateralize with fronto-parietal networks with related functions (e.g., language, response selection) (**Kanjlia et al., 2021**; **Lane et al., 2017**). For example, language-responsive occipital areas collateralize with language-responsive prefrontal areas across individuals (**Lane et al., 2017**). Recruitment of visual cortices by cross-modal tasks (e.g., spoken language) may enhance within-hemisphere connectivity in people born blind (**Kanjlia et al., 2021**; **Lane et al., 2017**; **Tian et al., 2022**). Together, this evidence supports the hypothesis that, starting from the less lateralized infant state, blindness increases lateralization of occipital long-range connectivity.

## Specialization of connectivity across different fronto-occipital networks: present in adults, absent at birth

In blind adults, different occipital areas show enhanced connectivity patterns with distinct subregions of PFC and this specialization is aligned with the functional specialization observed in task-based data (**Bedny et al., 2011**; **Kanjlia et al., 2016**; **Kanjlia et al., 2021**). For example, language-responsive

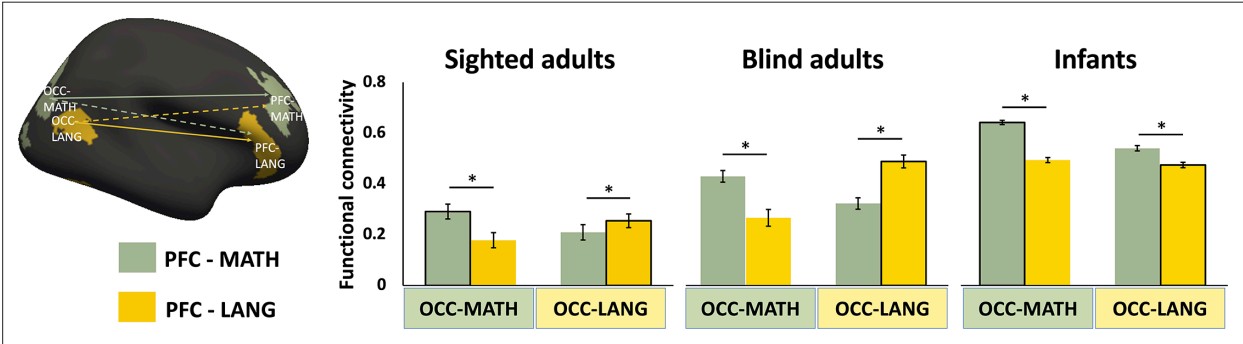

**Figure 4.** Occipito-frontal functional connectivity. The bar graph shows across-functional connectivity of different subregions of prefrontal (PFC) and occipital cortex (OCC) in sighted adults (n = 50), blind adults (n = 30), and infants (n = 475). Subregions (regions of interest) were defined based on task-based responses in a separate dataset of sighted (frontal) and blind (frontal and occipital) adults (**Kanjlia et al., 2016**; **Kanjlia et al., 2021**; **Lane et al., 2015**). PFC/OCC-MATH: math-responsive regions were more active when solving math equations than comprehending sentences. PFC/OC-LANG: language-responsive regions were more active when comprehending sentences than solving math equations (**Kanjlia et al., 2016**; **Kanjlia et al., 2021**; **Lane et al., 2015**). In blind adults, these regions show biases in connectivity related to their function, that is, language-responsive PFC is more correlated with language-responsive OCC. No such pattern is observed in infants. Asterisks (*) denote significant Bonferroni-corrected pairwise comparisons (p < 0.05, see Results section for statistical details). See **Figure 4—figure supplement 2** for connectivity matrix. Error bars represent SEM.

The online version of this article includes the following source data and figure supplement(s) for figure 4:

**Source data 1.** Post-hoc Bonferroni-corrected paired t-test for the connectivity between occipital regions to prefrontal regions in infants.

**Figure supplement 1.** Occipito-frontal functional connectivity across different subregions of prefrontal (PFC) and occipital cortex (OCC) insighted adults (n = 50), blind adults (n = 30), and infants (n = 475).

**Figure supplement 2.** The resting-state functional connectivity matrices between secondary visual areas to prefrontal regions in sighted adults (n = 50), blind adults (n = 30), and infants (n = 475).

subregions of occipital cortex show strongest functional connectivity with language-responsive subregions of PFC, whereas math-responsive occipital areas show stronger connectivity with math-responsive PFC. This pattern is most pronounced in blind people but can be seen weakly even in sighted participants (*Figure 4*; *Bedny et al., 2011*; *Kanjlia et al., 2016*; *Kanjlia et al., 2021*; *Lane et al., 2015*). Is this fronto-occipital connectivity specialization present in infancy, potentially enabling the task-based cross-modal specialization?

We compared connectivity preferences across three prefrontal and three occipital regions previously shown to activate preferentially in language (sentences > math), math (math > sentences), and response-conflict (no-go > go with tones) (*Kanjlia et al., 2016*; *Kanjlia et al., 2021*; *Lane et al., 2015*). For ease of viewing, *Figure 4* shows results from two of the three regions, math and language. See *Figure 4—figure supplement 1* for all three regions. Note that the statistical analyses included all three areas.

Contrary to the hypothesis that specialization of functional connectivity across different prefrontal/occipital areas is present from birth, infants showed a less differentiated fronto-occipital connectivity pattern relative to both blind and sighted adults (Group (sighted adults, blind adults, infants) by occipital regions (math, language, response-conflict) by PFC regions (math, language, response-conflict) interaction $F_{(8, 2208)} = 16.323$, $p < 0.001$). Unlike in adults, in infants, all the occipital regions showed stronger correlations with math- and response-conflict related prefrontal areas than language-responsive prefrontal areas (*Figure 4*, *Figure 4—figure supplement 1*). However, the preferential correlation with math-responsive PFC was strongest in those occipital areas that go on to develop math responses in blind adults (occipital regions (math, language, response-conflict) by PFC regions (math, language, response-conflict) interaction in infants $F_{(4, 1896)} = 85.145$, $p < 0.001$, post hoc Bonferroni-corrected paired *t*-test, see *Figure 4—source data 1*).

Note that findings of reduced regional specialization in infants need to be interpreted with caution. First, we do not know whether the same specialization of prefrontal subregions seen in adults is present in infants, although prior evidence suggests some prefrontal specialization is already present (*Raz and Saxe, 2020*). Second, the more fine-grained comparisons across occipital/frontal regions are more vulnerable to potential anatomical alignment issues between adult and infant brains. In other words, lack of specialization in infants could reflect the different location of the areas in this population.

## Discussion

The present results provide insight into the developmental process of experience-based functional specialization in human cortex. We find independent effects of visual experience and blindness on the development of visual networks. Aspects of the sighted adult connectivity pattern require visual experience. This is particularly striking for secondary visual cortices, where connectivity with non-visual networks in infants resembles blind more than sighted adults. Both in infants and blind adults, secondary occipital areas showed stronger functional connectivity with higher-order prefrontal cortices than with other sensory-motor networks (S1/M1, A1). Consistent with this observation, one previous study with a small sample of infants found strong connectivity between lateral occipital and prefrontal areas, although there was no comparison to blind adults in that study (*Barttfeld et al., 2018*). In V1, infants fell somewhere in between sighted and blind adults, suggesting an effect both of vision and of blindness on functional connectivity.

The present results reveal the effects of experience on development of functional connectivity between infancy and adulthood, but do not speak to the precise time course of these effects. Infants in the current sample had between 0 and 20 weeks of visual experience. Comparisons across these infants suggest that several weeks of postnatal visual experience is insufficient to produce a sighted adult connectivity profile. The time course of development could be anywhere between a few months and years and could be tested by examining data from children of different ages.

We propose that vision, as well as temporally coordinated multi-modal experiences, contributes to establishing the sighted adult connectivity profile in visual cortices. For example, coordinated visuo-motor activity during development may enhance connectivity between visual and motor networks. Supporting this conjecture, in infants, motor competence and early experience predict coupling between occipital and motor networks (*Colomer et al., 2023*).

Many questions remain regarding the neurobiological mechanisms underlying experience-based functional connectivity changes and their relationship to anatomical development. Resting-state functional correlations are an indirect measure of both function and anatomy, and differences in these measures are consistent with many possible underlying biological mechanisms. Long-range anatomical connections between brain regions are already present in infants—even prenatally—though they remain immature (*Huang et al., 2009*; *Kostović et al., 2021*; *Kostović et al., 2019*; *Takahashi et al., 2012*; *Vasung et al., 2017*). Functional connectivity changes may stem from local synaptic modifications within these stable structural pathways, consistent with findings that functional connectivity can vary independently of structural connection strength (*Fotiadis et al., 2024*). Moreover, functional connectivity has been shown to outperform structural connectivity in predicting individual behavioral differences, suggesting that experience-based functional changes may reflect finer-scale synaptic or network-level modulations not captured by macrostructural measures (*Ooi et al., 2022*). Prior studies also suggest that, even in adults, coordinated sensory-motor experience can lead to enhancement of functional connectivity across sensory-motor systems, indicating that large-scale changes in functional connectivity do not necessarily require corresponding changes in anatomical connectivity (*Guerra-Carrillo et al., 2014*; *Li et al., 2018*). Resting-state functional connectivity captures synchrony in blood oxygen level-dependent (BOLD) signal fluctuations rather than causal interactions, and differences in functional connectivity cannot on their own reveal how underlying neurophysiological mechanisms are modified. Connectivity changes between two areas could be mediated by 'third-party' hub regions. For example, posterior parietal cortex serves as a cortical hub for multisensory integration and visuo-motor coordination and could mediate occipital-to-sensory-motor communication (*Rolls et al., 2023*; *Sereno and Huang, 2014*). Subcortical structures such as the thalamus could also play a mediating role (*Vega-Zuniga et al., 2025*). Future studies will be needed to determine whether these functional changes are accompanied by alterations in structural connectivity and to probe causal interactions and mechanistic underpinnings.

The current findings reveal both effects of vision and effects of blindness on the functional connectivity patterns of the visual cortex. A further open question is whether visual experience plays an instructive or permissive role in shaping neural connectivity patterns. An instructive role implies that sensory experiences or patterns of neural activity directly shape and organize neural circuitry. In contrast, a permissive role implies that sensory experience or neural activity merely facilitates the influence of other factors—such as molecular signals—on the formation and organization of neural circuits (*Crair, 1999*; *Sur et al., 1999*). Studies with animals that manipulate the pattern or informational content of neural activity while keeping overall activity levels constant could distinguish between these hypotheses (*Crair, 1999*; *Roy et al., 2020*; *Stellwagen and Shatz, 2002*). In humans, such manipulations are not feasible, leaving us to study only the consequences of the presence or absence of vision. Under an instructive account, visual and multisensory experience could strengthen coupling between visual and other non-visual sensory-motor cortices through coordinated activity, thereby establishing the sighted adult connectivity pattern. In the absence of visual input, by contrast, the lack of such coordinated activity may prevent these couplings from being established. Alternatively, vision may act permissively, indirectly enabling maturational processes that shift connectivity toward the sighted adult configuration.

A further key question concerns the behavioral relevance of the connectivity signatures observed in the current study. The capacity of occipital cortices to support visual and multimodal behavior in sighted people may depend not only on local visual cortex function but also on the capacity of the visual system to coordinate its function with non-visual networks. Does enhanced connectivity between visual and non-visual sensory-motor networks facilitate multimodal integration for sighted people, for example, when catching a ball? Potentially consistent with this possibility, recent evidence suggests that people who grew up blind but recover sight in adulthood show multimodal integration deficits (*Badde et al., 2020*; *Guerreiro et al., 2015*; *Mowad et al., 2020*; *Putzar et al., 2007*) and distinct occipital oscillations (*Pant et al., 2023*).

Conversely, for people who remain blind throughout life, visual-PFC connectivity could enable recruitment of visual cortices for higher-order non-visual functions, such as language and executive control (*Bedny et al., 2011*; *Kanjlia et al., 2021*). Our results suggest that the pattern of connectivity observed in blind adults may build on connectivity patterns already present in infancy: like blind adults, infants show stronger occipital–PFC than occipital–sensory–motor coupling. Repeated

engagement of occipital networks during higher-cognitive tasks in early development could further enhance connectivity and enable specialization of visual networks for different non-visual higher-order functions.

Some prior studies have measured resting-state and task-based functional profiles in the same participants. These studies find that within visual cortices of blind people, the task-based profile of a cortical area is related to its resting-state connectivity pattern (*Abboud and Cohen, 2019*; *Deen et al., 2015*; *Kanjlia et al., 2016*; *Kanjlia et al., 2021*). This suggests that these two measures are related. However, the time course of this relationship, the developmental trajectory and mechanism of plasticity is not known. Primarily, this is because there is very little relevant developmental evidence. For example, in the current study, we find that the resting-state profile of secondary visual networks in infants is similar to that of blind adults. However, we do not know whether the visual cortices of infants show enhanced task-based cross-modal responses, relative to sighted adults, and how this compares to responses observed in blind adults. Future work with infants and children would be able to address this question.

In the current study, the clearest evidence for functional change driven by blindness was observed for laterality. Connectivity lateralization in sighted infants resembles that of sighted adults, in both V1 and secondary visual cortices. Relative to both sighted infants and sighted adults, blind adults show more lateralized connectivity patterns between occipital and prefrontal cortices. Previous studies suggest that in people born blind, occipital and non-occipital language responses are co-lateralized (*Lane et al., 2017*; *Tian et al., 2023*). We speculate that habitual activation of visual cortices by higher-cognitive tasks, such as language, which are themselves highly lateralized, contributes to this biased connectivity pattern of occipital cortex in blindness. Taken together, these results suggest a developmental framework in which intrinsic connectivity present in infancy provides a scaffold that is subsequently shaped and reinforced by experience-dependent recruitment, through either visual experience or the lifelong absence of vision in blindness. Longitudinal work across successive developmental stages will be crucial to test how the alternative trajectories shaped by visual experience versus blindness unfold over development.

## Materials and methods

### Participants

Fifty sighted adults and thirty congenitally blind adults contributed the resting-state data (sighted: $n$ = 50; 30 females; mean age = 35.33 years, standard deviation (SD) = 14.65; mean years of education = 17.08, SD = 3.1; blind: $n$ = 30; 19 females; mean age = 44.23 years, SD = 16.41; mean years of education = 17.08, SD = 2.11; blind vs. sighted age, $t_{(78)}$ = 2.512, p < 0.05; blind vs. sighted years of education, $t_{(78)}$ = 0.05, p = 0.996). Since blind participants were on average older, we also performed analyses in an age-matched subgroups of sighted controls ($n$ = 29) and found similar results to the full sample (see *Figure 1—figure supplement 6*). Blind and sighted participants had no known cognitive or neurological disabilities (screened through self-report). All adult anatomical images were read by a board-certified radiologist, and no gross neurological abnormalities were found. All the blind participants had at most minimal light perception from birth. Blindness was caused by pathology anterior to the optic chiasm (i.e., not due to brain damage). All participants gave written informed consent under a protocol approved by the Institutional Review Board of Johns Hopkins University.

Neonate data were from the third release of the dHCP ($n$ = 783) (https://www.developingconnectome.org). Ethical approval was obtained from the UK Health Research Authority (Research Ethics Committee reference number: 14/LO/1169). After quality control procedures (described below), 475 subjects were included in data analysis, with one scan per subject. The average age from birth at scan = 2.79 weeks (SD = 3.77, median = 1.57, range = 0–19.71); average gestational age at scan = 41.23 weeks (SD = 1.77, median = 41.29, range = 37–45.14); average gestational age at birth = 38.43 weeks (SD = 3.73, median = 39.71, range = 23–42.71). We only included infants who were full-term or scanned at term-equivalent age if preterm, while not being flagged by the dHCP project team as not passing quality control for fMRI images ($n$ = 634). Infants with more than 160 motion outliers were excluded ($n$ = 116 dropped). Motion-outlier volumes were defined as DVARS (the root mean square intensity difference between successive volumes) higher than 1.5 interquartile range above the 75th centile, after motion and distortion correction. Infants with signal drop-out in ROI were also

excluded (*n* = 43 dropped). To identify signal dropout, we first averaged BOLD signal intensity for all time points, for each subject, in each of 100 parcels defined by Schaefer's atlas (*Schaefer et al., 2018*). For each ROI (*n* = 18 ROIs) in the current study, signal dropout was then identified as BOLD intensity lower than –3 standard deviations, where the mean and standard deviations were identified across all 100 cortical parcels. Participants were excluded if any of the ROIs showed a signal dropout. The same signal dropout assessment was also applied to the blind and sighted adults to ensure consistent quality control across groups. One participant in the sighted adult group and two participants in the blind adult group exhibited signal dropout in one ROI each. Excluding these participants did not alter the group-level results (see *Figure 1—figure supplement 9*). The infants' structural images were reviewed by a pediatric neuroradiologist from the dHCP team, who assigned scores on a scale from 1 to 5. A score of 1 indicated a normal appearance for the subject's age, while scores of 4 or 5 suggested potential or likely clinical significance, or both clinical and imaging relevance. We repeated our analysis after excluding infants with a radiology score of 4 or 5, and the results remained consistent (see *Figure 1—figure supplement 7*).

## Image acquisition

### Blind and sighted adult

MRI anatomical and functional images were collected on a 3T Phillips scanner at the F. M. Kirby Research Center. T1-weighted anatomical images were collected using a magnetization-prepared rapid gradient-echo (MP-RAGE) in 150 axial slices with 1 mm isotropic voxels. Resting-state fMRI data were collected in 36 sequential ascending axial slices for 8 min. TR = 2 s, TE = 0.03 s, flip angle = 70°, voxel size = 2.4 × 2.4 × 2.5 mm, inter-slice gap = 0.5 mm, field of view = 192 × 172.8 × 107.5. Participants completed 1–4 scans of 240 volume each (average scan time = 710.4 s per person). During the resting-state scan, participants were instructed to relax but remain awake. Sighted participants wore light-excluding blindfolds to equalize the light conditions across the groups during the scans.

### Infants (dHCP)

Anatomical and functional images were collected on a 3T Phillips scanner at the Evelina Newborn Imaging Centre, St Thomas' Hospital, London, UK. A dedicated neonatal imaging 219 system including a neonatal 32-channel phased-array head coil was used. T2w multi-slice fast spin-echo images were acquired with in-plane resolution 0.8 × 0.8 mm$^2$ and 1.6 mm slices overlapped by 0.8 mm (TR = 12,000 ms, TE = 156 ms, SENSE factor 2.11 axial and 2.6 sagittal). In infants, T2w images were used as the anatomical image because the brain anatomy is more clearly in T2w than in T1w images. Fifteen minutes of resting-state fMRI data were collected using a used multi-band 9x accelerated echo-planar imaging (TR = 392 ms, TE = 38 ms, 2300 volumes, with an acquired resolution of 2.15 mm isotropic). Single-band reference scans were acquired with bandwidth-matched readout, along with additional spin-echo acquisitions with both AP/PA fold-over encoding directions.

## Data analysis

Resting-state data were preprocessed using FSL version 5.0.9 (*Smith et al., 2004*), DPABI version 6.1 (*Yan et al., 2016*), FreeSurfer (*Dale et al., 1999*), and in-house code (https://github.com/NPDL/Resting-state_dHCP, copy archived at *Tian, 2023*). The functional data for all groups were linearly detrended and low-pass filtered (0.08 Hz).

For adults, functional images were registered to the T1-weighted structural images, motion corrected using MCFLIRT (*Jenkinson et al., 2002*), and temporally high-pass filtering (150 s). No subject had excessive head movement (>2 mm) or rotation (>2°) at any timepoint. Resting-state data are known to include artifacts related to physiological fluctuations such as cardiac pulsations and respiratory-induced modulation of the main magnetic field. A component-based method, CompCor (*Behzadi et al., 2007*), was therefore used to control for these artifacts. Particularly, following the procedure described in Whitfield-Gabrieli et al., nuisance signals were extracted from two-voxel eroded masks of spinal fluid (CSF) and white matter (WM), and the first five principal components analysis components derived from these signals was regressed out from the processed BOLD time series (*Whitfield-Gabrieli and Nieto-Castanon, 2012*). In addition, a scrubbing procedure was applied to further reduce the effect of motion on functional connectivity measures (*Power et al., 2012*; *Power*

*et al., 2014*). Frames with root mean square intensity difference exceeding 1.5 interquartile range above the 75th centile, after motion and distortion correction, were censored as outliers.

The infants' resting-state functional data were preprocessed by the dHCP group using the project's in-house pipeline (*Fitzgibbon et al., 2020*). This pipeline uses a spatial independent component analysis (ICA) denoising step to minimize artifact due to multi-band artifact, residual head movement, arteries, sagittal sinus, CSF pulsation. For infants, ICA denoising is preferable to using CSF/WM regressors. Because it is challenging to accurately define anatomical boundaries of CSF/WM due to the low imaging resolution compared with the brain size and the severe partial-volume effect in the neonate (*Fitzgibbon et al., 2020*). Like in the adults, frames with root mean square intensity difference exceeding 1.5 interquartile range above the 75th centile, after motion and distortion correction, were considered as motion outliers. Out of the 2300 frames, a subset of continuous 1600 with a minimum number of motion outliers was kept for each subject. Motion outliers were censored from the subset of continuous 1600, and a subject was excluded from further analyses when the number of outliers exceeded 160 (10% of the continuous subset) (*Hu et al., 2022*). While infant connectivity estimates may be less robust at the individual level compared to adults due to shorter scan durations and higher motion, our cohort's large sample size ($n = 475$) and rigorous motion censoring mitigate these limitations for group-level analyses. Substantial differences between the groups exist in this study, including the number of subjects, brain sizes, imaging parameters, and data preprocessing, all of which are likely to have an impact on the overall signal quality. To assess the reliability of the ROI-wise FC used in all analyses, we computed split-half noise ceilings for each participant. The rs-fMRI time series was divided into two equal halves, ROI-wise FC profiles were calculated separately for each half, and the noise ceiling for each ROI was defined as the Pearson correlation between the two profiles following *Lage-Castellanos et al., 2019*. We then averaged the ROI-wise noise ceilings for each group. The resulting values (infants: 0.90 ± 0.056; blind adults: 0.88 ± 0.041; sighted adults: 0.90 ± 0.055) did not differ significantly (one-way ANOVA: $F_{(2,552)} = 2.348$, p = 0.097). We also examined noise ceilings for each ROI separately. All ROIs showed high absolute reliability (noise ceiling >0.80) across groups. Although many ROIs showed statistically significant group differences in noise ceiling (one-way ANOVA, p < 0.05), the effect sizes were small to moderate (partial $\eta^2 < 0.14$). These findings indicate that reliability may vary modestly across groups at the ROI level. We cannot determine whether such variability contributes to the group differences reported in this study, but the consistently high absolute reliability suggests that the FC estimates are generally robust. The full ROI-wise statistical results are provided in *Supplementary file 1*. For both groups of adults and infants, we performed a temporal low-pass filter (0.08 Hz low-pass cutoff) and a linear detrending. ROI-to-ROI connectivity was calculated using Pearson's correlation between ROI-averaged BOLD time series (ROI definition see below). The all *t*-tests and *F*-tests are two-sided. The comparison of correlation coefficients was done using the cocor software package and Pearson and Filon's *z* (*Diedenhofen and Musch, 2015*; *Pearson and Filon, 1898*).

## ROI definition

Frontal and secondary visual ROIs were defined functionally based on data from a separate task-based fMRI experiments with blind and sighted adults (*Kanjlia et al., 2016*; *Kanjlia et al., 2021*; *Lane et al., 2015*). Three separate experiments were conducted with the same group of blind and sighted subjects (sighted $n = 18$; blind $n = 23$). The language ROIs in the occipital and frontal cortices were identified by sentence > nonwords contrast in an auditory language comprehension task (*Lane et al., 2015*). The math ROIs were identified by math > sentence contrast in an auditory task where participants judged equivalence of pairs of math equations and pairs of sentences (*Kanjlia et al., 2016*). The executive function ROIs were identified by no-go > frequent go contrast in an auditory go/no-go task with non-verbal sounds (*Kanjlia et al., 2021*). All ROI files are available at openICPSR (https://doi.org/10.3886/E198832V1).

Occipital secondary 'visual' ROIs were defined based on group comparisons blind > sighted in a whole-cortex analysis (*Figure 1*, *Figure 1—figure supplement 5*.) The occipital language ROI was defined as the cluster that responded more to auditory sentence than auditory nonwords conditions in blind, relative to sighted, in a whole-cortex analysis, likewise the occipital math ROI was defined as math > sentences, blind > sighted interaction and the occipital executive ROI as no-go > frequent go, blind > sighted. All three occipital ROIs were defined in the right hemisphere. Left-hemisphere occipital

ROIs were created by flipping the right-hemisphere ROIs to the left hemisphere. Each functional ROI spans multiple anatomical regions, and together the secondary visual ROIs tile large portions of lateral occipital, occipito-temporal, dorsal occipital, and occipito-parietal cortices. In sighted people, the secondary visual occipital ROIs include the anatomical locations of functional regions such as motion area V5/MT+, the lateral occipital complex (LO), as well as ventral portions encompassing category-selective ventral occipito-temporal cortices, including V4v, and dorsal portions including V3a. The occipital ROI also extended ventrally into the middle portion of the temporal lobe and dorsally into the intraparietal sulcus and superior parietal lobule. The frontal PFC ROIs were defined functionally, based on a whole-cortex analysis which combined all blind and sighted adult data. The frontal language ROI was defined as responding more to auditory sentence than auditory nonword conditions across all blind and sighted subjects, constrained to the prefrontal cortex. Likewise, math-responsive PFC was defined as math > sentences and executive no-go > frequent go. For frontal ROIs, the language ROI was defined in the left, and the math and executive function ROI were defined in the right hemisphere, then flipped to the other hemisphere.

The V1 ROI was defined from a previously published anatomical surface-based atlas (PALS-B12) (*Van Essen, 2005*). The primary somatosensory and motor cortex (S1/M1) ROI was selected as the area that responds more to the go than no-go trials in the auditory go/no-go task across both blind and sighted groups, constrained to the hand area in S1/M1 search space from neurosynth.org (term 'hand movements') (*Kanjlia et al., 2021*). The primary auditory cortex (A1) ROI was defined as the transverse temporal portion of a gyral-based atlas (*Desikan et al., 2006*; *Morosan et al., 2001*).

All the ROIs were defined in standard space and then transformed into each subject's native space. For adults, this was done by employing the deformation field estimated by FreeSurfer. For infants, the ROIs were transformed into each subject's native space using a two-step approach. First, the ROIs were converted from the adult's MNI space into the 40-week dHCP template (*Bozek et al., 2018*). ANTS, previously shown to be effective in pediatric studies (*Avants et al., 2014*; *Cabral et al., 2022*; *Jain et al., 2012*; *Lawson et al., 2013*), was utilized to estimate the deformation field between these two spaces. In this step, the infant's scalp and cerebellum were masked, as these structures in the infant brain greatly differ from those in the adult and can introduce bias into the registration process, as outlined in a study by *Cabral et al., 2022*. Second, the ROIs were further transformed from the 40-week template space into each individual's native spaces, employing the deformation field provided by the dHCP group. Nearest neighbor interpolation was applied in both steps (examples of the resulting ROI alignment on individual brains are shown in *Figure 1—figure supplement 8*). For both adults and infants, any overlapping voxels between ROIs were removed and not counted toward any ROIs.

## Acknowledgements

We would like to thank all the blind and sighted participants, the blind community, and the National Federation of the Blind. Without their support, this study would not be possible. We would also like to thank the FM Kirby Research Center for Functional Brain Imaging at the Kennedy Krieger Institute for their assistance in data collection. Xiang Xiao was supported by the Intramural Research Program of the National Institute on Drug Abuse, the National Institute of Health, United States. This work was supported by grants from the National Eye Institute at the National Institutes of Health (R01EY027352-01 and R01EY033340). RC was supported by the ERC Advanced Grant 'Foundations of Cognition' (FOUNDCOG) 787981. MT was supported by grants from the National Natural Science Foundation of China (32400891). XX was supported by grants from the National Natural Science Foundation of China (32400844) and the Fundamental Research Funds for the Central Universities.

## Additional information

### Funding

| Funder | Grant reference number | Author |
| --- | --- | --- |
| National Eye Institute | R01EY027352-01 | Marina Bedny |

| Funder | Grant reference number | Author |
|---|---|---|
| National Eye Institute | R01EY033340 | Marina Bedny |
| ERC Advanced Grant "Foundations of Cognition" | 787981 | Rhodri Cusack |
| National Natural Science Foundation of China | 32400891 | Mengyu Tian |
| National Natural Science Foundation of China | 32400844 | Xiang Xiao |
| The Fundamental Research Funds for the Central Universities | | Mengyu Tian |
| The Fundamental Research Funds for the Central Universities | | Xiang Xiao |

The funders had no role in study design, data collection, and interpretation, or the decision to submit the work for publication.

## Author contributions

Mengyu Tian, Conceptualization, Data curation, Formal analysis, Visualization, Writing – original draft, Writing – review and editing; Xiang Xiao, Data curation, Software, Formal analysis, Visualization, Methodology, Writing – review and editing; Huiqing Hu, Software, Methodology; Rhodri Cusack, Supervision, Methodology, Writing – review and editing; Marina Bedny, Conceptualization, Supervision, Funding acquisition, Writing – original draft, Writing – review and editing

## Author ORCIDs

Mengyu Tian (ID) https://orcid.org/0000-0003-2289-4415
Xiang Xiao (ID) https://orcid.org/0000-0002-2266-4284
Huiqing Hu (ID) https://orcid.org/0000-0001-5652-9606
Rhodri Cusack (ID) https://orcid.org/0000-0002-5234-7415

## Ethics

All the blind and sighted adult participants gave written informed consent under a protocol approved by the Institutional Review Board of Johns Hopkins University. The infant data were obtained from the Developing Human Connectome Project, with ethical approval and informed consent provided by the original investigators.

Reviewer #1 (Public review): https://doi.org/10.7554/eLife.93067.4.sa1
Reviewer #2 (Public review): https://doi.org/10.7554/eLife.93067.4.sa2
Reviewer #3 (Public review): https://doi.org/10.7554/eLife.93067.4.sa3
Author response https://doi.org/10.7554/eLife.93067.4.sa4

# Additional files

## Supplementary files

Supplementary file 1. ROI-wise noise ceiling statistics across groups.
MDAR checklist

## Data availability

Neonate data were from the third release of the Developing Human Connectome Project (https://www.developingconnectome.org). The de-identified blind and sighted adults' data were posted on openICPSR (https://doi.org/10.3886/E198832V1).

The following dataset was generated:

| Author(s) | Year | Dataset title | Dataset URL | Database and Identifier |
|---|---|---|---|---|
| Bedny M, Tian M | 2023 | Blindness Resting State | https://doi.org/10.3886/E198832V1 | OPENICPSR, 10.3886/E198832V1 |

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
