## [Editor Report · eLife Assessment]

This **important** study provides evidence supporting the hypothesis that postnatal visual experience shapes the patterns of functional connectivity between extrastriate visual cortex and frontal regions, by comparing neonates, blind and sighted adults using resting-state fMRI. The evidence supporting the main claim is **convincing**, and the authors' interpretations are appropriately calibrated in the discussion. Nevertheless, the study design and methodology are inherently limited to resolve the underlying mechanisms driving connectivity changes during neurodevelopment (experience-related plasticity vs post-natal experience-independent maturation). This study will be of broad interest to neuroscientists and neuroimaging researchers studying vision, plasticity and brain development.

---

## [Referee Report · Reviewer #1 (Public review)]

Summary:

The present study evaluates the role of visual experience in shaping functional correlations between human extrastriate visual cortex and frontal regions. The authors used fMRI to assess "resting-state" temporal correlations in three groups: sighted adults, congenitally blind adults, and neonates. Previous research has already demonstrated differences in functional correlations between visual and frontal regions in sighted compared to early blind individuals. The novel contribution of the current study lies in the inclusion of an infant dataset, which allows for an assessment of the developmental origins of these differences.

The main results of the study reveal that correlations between prefrontal and visual regions are more prominent in the blind and infant groups, with the blind group exhibiting greater lateralization. Conversely, correlations between visual and somato-motor cortices are more prominent in sighted adults. Based on these data, the authors conclude that visual experience shapes these cortical networks through activity-dependent plasticity. This study provides novel insights into the impact of visual experience on the development of temporal correlations in the brain.

Strengths:

The dissociations in functional correlations observed among the sighted adult, congenitally blind, and neonate groups provide strong support for the main conclusion regarding postnatal experience-driven shaping of visual-frontal connectivity.

The neonatal data offers a unique and valuable developmental anchor for interpreting divergence between blind and sighted adults. This is a major advance over prior studies limited to adult comparisons.

Convergence with prior findings in the blind and sighted adult groups reinforces the reliability and external validity of the present results.

The split-half reliability analysis in the infant and adult data increases confidence in the robustness of the reported group differences.

Weaknesses:

The methodology cannot determine whether group differences in correlations reflect direct changes in communication between visual and frontal regions or indirect effects mediated by other structures.

The cross-sectional design cannot reveal the timecourse over which visual experience shapes connectivity between infancy and adulthood.

Whether the infant resting-state patterns imply similar functional capacity to blind adults (e.g., cross-modal task responses) remains untested.

Comments on revisions:

The authors have done a fantastic job addressing my remaining questions.

---

## [Referee Report · Reviewer #2 (Public review)]

Summary:

Tian et al. explore the developmental origins of cortical reorganization in blindness. Previous work has found that a set of regions in the occipital cortex show different functional responses and patterns of functional correlations in blind vs. sighted adults. Here, Tian et al. explore how this organisation arises over development, asking whether the infant brain looks more like the blind adult pattern, or more like the sighted adult pattern. Their analyses reveal that the answer depends on the particular networks investigated. Some functional connections in infants look more like blind than sighted adults; other functional connections look more like sighted than blind adults; and others fall somewhere in the middle, or show an altogether different pattern in infants compared with both sighted and blind adults.

Strengths:

The paper addresses very important questions about the "starting state" in the developing visual cortex, and how cortical networks are shaped by experience. Another clear strength lies in the unequivocal nature of many results. Many results have very large effect sizes, critical interactions between regions and groups are tested and found, and infant analyses are replicated in split halves of the data.

Weaknesses:

While potential roles of experience (e.g., visual, cross-modal) are discussed in detail, little consideration is given to the role of experience-independent maturation. The infants scanned are extremely young, only 2 weeks old. It is possible that the sighted adult pattern may still emerge later in infancy or childhood, regardless of infant visual experience. If so, the blind adult pattern may depend on blindness-related experience only (which may or may not reflect "visual" experience per se). In short, it is not clear that the age range studied is a clear-cut "starting point" for development, after which all change can be attributed to experience.

---

## [Referee Report · Reviewer #3 (Public review)]

Summary

This study aimed to investigate whether the differences observed in the organization of visual brain networks between blind and sighted adults result from a reorganization of an early functional architecture due to blindness, or whether the early architecture is immature at birth and requires visual experience to develop functional connections. This question was investigated through the comparison of 3 groups of subjects with resting-state functional MRI (rs-fMRI). Based on convincing analyses, the study suggests that: (1) secondary visual cortices showed higher connectivity to prefrontal cortical regions (PFC) than to non-visual sensory areas (S1/M1 and A1) in infants like in blind adults, in contrast to sighted adults; (2) the V1 connectivity pattern of infants lies between that of sighted adults (showing stronger functional connectivity with non-visual sensory areas than with PFC) and that of blind adults (showing stronger functional connectivity with PFC than with non-visual sensory areas); (3) the laterality of the connectivity patterns of infants resembled those of sighted adults more than those of blind adults, but infants showed a less differentiated fronto-occipital connectivity pattern than adults.

Strengths

- The question investigated in this article is important for understanding the mechanisms of plasticity during typical and impaired development, and the approach considered, which compares different groups of subjects including, neonates/infants and blind adults, is highly original.

- Overall, the presented analyses are solid and well-detailed, and the results and discussion are convincing.

Weaknesses

- While it is informative to compare the "initial" state (close to birth) and the "final" states in blind and sighted adults to study the impact of post-natal and visual experience, this study does not analyze the chronology of this development and when the specialization of functional connections is completed. This would require investigating the evolution of functional connectivity of the visual system as a function of visual experience and thus as a function of age, at least during toddlerhood given the early and intense maturation of the visual system after birth. This could be achieved by analyzing different developmental periods using open databases such as the Baby Connectome Project.

- The rationale for grouping full-term neonates and preterm infants (scanned at term-equivalent age) is not understandable when seeking to perform comparisons with adults. Even if the study results do not show differences between full-terms and preterms in terms of functional connectivity differences between regions and of connectivity patterns, preterms group had different neurodevelopment and post-natal (including visual) experiences (even a few weeks might have an impact). And actually they show reduced connectivity strength systematically for all regions compared with full-terms (Sup Fig 7). Considering a more homogeneous group of neonates would have strengthened the study design.

- The rationale for presenting results on the connectivity of secondary visual cortices before the one of primary cortices (V1) could be clarified.

- The authors acknowledge the methodological difficulties for defining regions of interest (ROIs) in infants in a similar way as adults. Since the brain development is not homogeneous and synchronous across brain regions (in particular with the frontal and parietal lobes showing a delayed growth), this poses major problems for registration. This raises the question of whether the study findings could be biased by differences in ROI positioning across groups.

Comments on revisions:

The authors have addressed my specific recommendations, but some weaknesses in the study remain, particularly the inclusion of preterm infants alongside full-term neonates.

---

## [Author Response]

The following is the authors’ response to the previous reviews

**Public Reviews:**

**Reviewer #1 (Public review):**
Summary:The present study evaluates the role of visual experience in shaping functional correlations between human extrastriate visual cortex and frontal regions. The authors used fMRI to assess "resting-state" temporal correlations in three groups: sighted adults, congenitally blind adults, and neonates. Previous research has already demonstrated differences in functional correlations between visual and frontal regions in sighted compared to early blind individuals. The novel contribution of the current study lies in the inclusion of an infant dataset, which allows for an assessment of the developmental origins of these differences.The main results of the study reveal that correlations between prefrontal and visual regions are more prominent in the blind and infant groups, with the blind group exhibiting greater lateralization. Conversely, correlations between visual and somato-motor cortices are more prominent in sighted adults. Based on these data, the authors conclude that visual experience plays an instructive role in shaping these cortical networks. This study provides valuable insights into the impact of visual experience on the development of functional connectivity in the brain.Strengths:The dissociations in functional correlations observed among the sighted adult, congenitally blind, and neonate groups provide strong support for the main conclusion regarding postnatal experience-driven shaping of visual-frontal connectivity.The inclusion of neonates offers a unique and valuable developmental anchor for interpreting divergence between blind and sighted adults. This is a major advance over prior studies limited to adult comparisons.Convergence with prior findings in the blind and sighted adult groups reinforces the reliability and external validity of the present results.The split-half reliability analysis in the infant data increases confidence in the robustness of the reported group differences.Weaknesses:The manuscript risks overstating a mechanistic distinction between sighted and blind development by framing visual experience as "instructive" and blindness as "reorganizing." Similarly, the binary framing of visual experience and blindness as independent may oversimplify shared plasticity mechanisms.The interpretation of changes in temporal correlations as altered neural communication does not adequately consider how shifts in shared variance across networks may influence these measures without reflecting true biological reorganization.The discussion does not substantively engage with the longstanding debate over whether sensory experience plays an instructive or permissive role in cortical development.The relationship between resting-state and task-based findings in blindness remains unclear.
**Reviewer #2 (Public review):**
Summary:Tian et al. explore the developmental origins of cortical reorganization in blindness. Previous work has found that a set of regions in the occipital cortex show different functional responses and patterns of functional correlations in blind vs. sighted adults. Here, Tian et al. explore how this organization arises over development. Is the "starting state" more like the blind pattern, or more like the adult pattern? Their analyses reveal that the answer depends on the particular networks investigated. Some functional connections in infants look more like blind than sighted adults; other functional connections look more like sighted than blind adults; and others fall somewhere in the middle, or show an altogether different pattern in infants compared with both sighted and blind adults.Strengths:The paper addresses very important questions about the starting state in the developing visual cortex, and how cortical networks are shaped by experience. Another clear strength lies in the unequivocal nature of many results. Many results have very large effect sizes, critical interactions between regions and groups are tested and found, and infant analyses are replicated in split halves of the data.Weaknesses:While potential roles of experience (e.g., visual, cross-modal) are discussed in detail, little consideration is given to the role of experience-independent maturation. The infants scanned are extremely young, only 2 weeks old. It is possible then that the sighted adult pattern may still emerge later in infancy or childhood, regardless of infant visual experience. If so, the blind adult pattern may depend on blindness-related experience only (which may or may not reflect "visual" experience per se). In short, it is not clear that birth, or the first couple weeks of life, are a clear cut "starting point" for development, after which all change can be attributed to experience.
**Reviewer #3 (Public review):**
SummaryThis study aimed to investigate whether the differences observed in the organization of visual brain networks between blind and sighted adults result from a reorganization of an early functional architecture due to blindness, or whether the early architecture is immature at birth and requires visual experience to develop functional connections. This question was investigated through the comparison of 3 groups of subjects with resting-state functional MRI (rs-fMRI). Based on convincing analyses, the study suggests that: (1) secondary visual cortices showed higher connectivity to prefrontal cortical regions (PFC) than to non-visual sensory areas (S1/M1 and A1) in infants like in blind adults, in contrast to sighted adults; (2) the V1 connectivity pattern of infants lies between that of sighted adults (showing stronger functional connectivity with non-visual sensory areas than with PFC) and that of blind adults (showing stronger functional connectivity with PFC than with non-visual sensory areas); (3) the laterality of the connectivity patterns of infants resembled those of sighted adults more than those of blind adults, but infants showed a less differentiated fronto-occipital connectivity pattern than adults.Strengths- The question investigated in this article is important for understanding the mechanisms of plasticity during typical and impaired development, and the approach considered, which compares different groups of subjects including, neonates/infants and blind adults, is highly original.- Overall, the presented analyses are solid and well detailed, and the results and discussion are convincing.Weaknesses- While it is informative to compare the "initial" state (close to birth) and the "final" states in blind and sighted adults to study the impact of post-natal and visual experience, this study does not analyze the chronology of this development and when the specialization of functional connections is completed. This would require investigating the evolution of functional connectivity of the visual system as a function of visual experience and thus as a function of age, at least during toddlerhood given the early and intense maturation of the visual system after birth. This could be achieved by analyzing different developmental periods using open databases such as the Baby Connectome Project.- The rationale for grouping full-term neonates and preterm infants (scanned at term-equivalent age) is not understandable when seeking to perform comparisons with adults. Even if the study results do not show differences between full-terms and preterms in terms of functional connectivity differences between regions and of connectivity patterns, preterms group had different neurodevelopment and post-natal (including visual) experiences (even a few weeks might have an impact). And actually they show reduced connectivity strength systematically for all regions compared with full-terms (Sup Fig 7). Considering a more homogeneous group of neonates would have strengthen the study design.- The rationale for presenting results on the connectivity of secondary visual cortices before the one of primary cortices (V1) could be clarified.- The authors acknowledge the methodological difficulties for defining regions of interest (ROIs) in infants in a similar way as adults. Since the brain development is not homogeneous and synchronous across brain regions (in particular with the frontal and parietal lobes showing a delayed growth), this poses major problems for registration. This raises the question of whether the study findings could be biased by differences in ROI positioning across groups.
**Recommendations for the authors:**

**Reviewer #1 (Recommendations for the authors):**
The authors are appropriately cautious in many parts of the discussion and include several helpful control analyses. Nonetheless, additional clarification of key assumptions and potential confounds would strengthen the paper.(1) The current framing labels vision as "instructive" and blindness as "reorganizing," but it is unclear why these two experiential factors are characterized differently. Both involve activity-dependent changes to functional architecture from a shared immature scaffold. Labeling them differently risks conflating divergent outcomes with distinct underlying mechanisms. Just because visual and blind adults show different patterns of functional connectivity does not mean they reflect separate processes. While the discussion briefly acknowledges the possibility of shared plasticity mechanisms, much of the framing across the manuscript, including in the abstract and introduction, implies a dichotomy. A clearer articulation of the criteria used to assign these labels, or reconsideration of whether such a distinction is warranted, would improve conceptual clarity. The current framing appears analogous to saying that "heat causes expansion" and "cold causes contraction" as if these were separate mechanisms, when they are actually two directions of change along a single factor: temperature. A more parsimonious framework, such as activity-dependent reweighting of pre-existing connectivity, may better capture the nature of plasticity at play in both sighted and blind development.

Following the reviewer’s suggestion, we have revised the manuscript to clarify that both vision and blindness can be understood as manifestations of a common framework of experience-driven plasticity. We removed all mention of reorganization and clarify and modified the wording throughout.

Specifically:

Abstract: “Are infant visual cortices functionally like those of sighted adults, with blindness leading to functional change? We find that, on the contrary that secondary visual cortices of infants are functionally more like those of blind adults: stronger coupling with PFC than with nonvisual sensory-motor networks, suggesting that visual experience modifies elements of the sighted-adult long-range functional connectivity profile. Infant primary visual cortices are in-between blind and sighted adults i.e., more balanced PFC and sensory-motor connectivity than either adult group. The lateralization of occipital-to-frontal connectivity in infants resembles the sighted adults, consistent with the idea that blindness leads to functional change. These results suggest that both vision and blindness modify functional connectivity through experience-driven (i.e., activity-dependent) plasticity.” (Page 1, Line 13)

Introduction: We replaced “blindness leads to functional reorganization” with “blindness modifies this functional connectivity” (Page 2, Line 52), and the following sentence has also been modified to: “lifetime visual experience shapes connectivity toward the sighted-adult pattern” (Page 2, Line 54) For the lateralization patterns, we now describe them as “blindness-related modification” rather than “reorganization”, to keep the interpretation descriptive rather than mechanistic. (Page 4, Line 114),

(2) In interpreting the functional correlation differences, the discussion should more explicitly consider how statistical interdependence between areas could influence the observed results. For example, an increase in shared variance between visual and motor areas, such as might result from visually guided action, could result in a reduction in the apparent strength of visual-prefrontal temporal correlation (at the resolution of fMRI) without any true biological change in communication between visual-prefrontal cortex. This possibility is not ruled out by reporting groupwise patterns of relative connectivity. A more cautious systems-level framing could help clarify the distinction between neural plasticity and statistical redistribution of variance.

We thank the reviewer for raising this important point. We agree that resting-state fMRI provides a measure of statistical synchrony in BOLD signals rather than direct causal interactions between regions. This a fundamental limitation of resting state fMRI, which we now note in the Discussion section. Such changes in correlation are consistent with a variety of underlying biological mechanisms. Online task is one factor that influences cross-region correlations. In the current study, both blind and sighted groups were measured while blindfolded and were not performing visually guided actions during the resting state fMRI scans. It is possible that past visual-guided action *experience* changes the resting state correlations of sighted participants. Indeed, this is one interesting hypothesis.

In the revised Discussion, we now explicitly note this limitation and clarify that differences in FC do not by themselves establish whether or how underlying neurophysiological mechanisms are changed. We also emphasize that future work will need to investigate whether FC changes are accompanied by alterations in structural connectivity and to probe causal interactions and mechanistic underpinnings as follows：

“Resting-state functional connectivity captures synchrony in BOLD signal fluctuations rather than causal interactions and differences in functional connectivity cannot on their own reveal how underlying neurophysiological mechanisms are modified.” (page 13,line 342)

“Future studies will be needed to determine whether these functional changes are accompanied by alterations in structural connectivity, and to probe causal interactions and mechanistic underpinnings.” (page 13,line 350)

(3) The mechanistic interpretation of group differences in visual-motor coupling would benefit from stronger network-level justification. Direct connections between these areas are sparse in primates. If effects reflect indirect polysynaptic interactions or shared thalamic input, as the authors suggest, one might expect corresponding group differences in intermediate regions (e.g., parietal cortex, thalamus) that mediate these interactions. Is there any evidence for this in the data?

We thank the reviewer for raising this point. We agree and as noted above, resting state fMRI cannot distinguish between direct causal interactions between two regions and ones that a mediating region is involved. This is a fundamental limitation of resting state fMRI. The current study further focused on testing a specific hypothesis motivated by previously observed group differences between blind and sighted adults and our analyses focused on ROI-to-ROI connectivity between occipital, frontal, and sensory-motor cortices, and did not include these additional regions. In prior work, we and others, have looked at effects in parietal cortices (Abboud & Cohen, 2019; Bedny et al., 2009; Deen et al., 2015; Kanjlia et al., 2016, 2021; Sen et al., 2022). In blindness, parietal networks show increased correlations with some visual areas, rather than decreased. Regarding the thalamus, there is less clear evidence and there is some ongoing work trying to address this question. A couple of studies suggest that there is indeed increased connectivity between some parts of the thalamus and visual cortex in blindness. Although the anatomical information is limited, some of the work suggests that this increase is with higher-cognitive nuclei of the thalamus (Bedny et al., 2011; Liu et al., 2007).

We agree that this is an important direction for future work. To acknowledge this point, we have revised the manuscript to highlight the potential role of cortical and subcortical hub regions in mediating connectivity changes. The text has been modified as follows:

“Connectivity changes between two areas could be mediated by ‘third-party’ hub regions. For example, posterior parietal cortex serves as a cortical hub for multisensory integration and visuo-motor coordination and could mediate occipital-to-sensory-motor communication (Rolls et al., 2023; Sereno & Huang, 2014). Subcortical structures such as the thalamus could also play a mediating role (Vega-Zuniga et al., 2025).” (page 13,line 345)

(4) The discussion would benefit from deeper engagement with prior work on experience-dependent plasticity, particularly the longstanding distinction between instructive and permissive roles of experience. While the authors briefly define these concepts and reference their historical use, a more explicit consideration of how their findings relate to this broader literature would help clarify whether such distinctions are necessary or appropriate.

We thank the reviewer for this thoughtful suggestion to engage more explicitly with the longstanding literature on instructive versus permissive roles of experience. However, most of this literature comes from animal models, where experimental manipulations of the anatomical structure, of experience itself (e.g., controlled rearing studies) and sometimes of neural activity patterns allow clear tests of these mechanisms. Such manipulations are not feasible in humans. The terminology in the animal literature does not directly map onto the methods and data available in the present study or in other work with humans. For this reason, the current data does not allow us to fully engage with the debates in the animal literature and doing risks overinterpreting our findings.

Nevertheless, we agree that once the instructive/permissive framework has been introduced, it is important to clarify how our results relate to it, rather than only providing definitions. We have therefore added the following text to the discussion:

“In humans, such manipulations are not feasible, leaving us to study only the consequences of the presence or absence of vision. Under an instructive account, visual and multisensory experience could strengthen coupling between visual and other non-visual sensory-motor cortices through coordinated activity, thereby establishing the sighted-adult connectivity pattern. In the absence of visual input, by contrast, the lack of such coordinated activity may prevent these couplings from being established. Alternatively, vision may act permissively, indirectly enabling maturational processes that shift connectivity toward the sighted-adult configuration.” (page 14,line 362)

(5) The revised discussion acknowledges the divergence between resting-state and task-based findings, but does not fully frame the theoretical implications of this discrepancy. Although this study cannot resolve the issue with its own data, a more integrative discussion could help clarify whether these measures reflect distinct functional states, developmental trajectories, or mechanisms of plasticity. Without such framing, readers are left without clear guidance on how to reconcile the present results with prior work on cross-modal recruitment in blindness.

We thank the reviewer for this thoughtful comment. We agree that know how resting-state evidence relates to task-based evidence is a fundamentally important issue. We now discuss this more in the Introduction as well as in the Discussion.

There is a sizable literature of both task-based and resting state studies. Some of prior studies have measured resting state and task-based data within the same participants and found relationships (Kanjlia et al., 2016, 2021; Lane et al., 2015). We now clarify this in the introduction. These studies find that within visual cortices of blind people, the task-based profile of a cortical area is related to its resting state connectivity pattern (Abboud & Cohen, 2019; Deen et al., 2015; Kanjlia et al., 2016, 2021). This suggests that these two measures are related. However, the timecourse of this relationship, the developmental trajectory and mechanism of plasticity is not known. We note this now in the introduction on page 2. Primarily this is because there is very little relevant developmental evidence. For example, in the current study we find that the resting state profile of secondary visual networks in infants is similar to that of blind adults. However, we do not know whether the visual cortices of infants show task-based cross modal responses. To our knowledge nobody has tested this question. We agree with the reviewer that raising this question in the paper is better than not commenting on the relationship at all.

To address the reviewer’s comment, we have expanded the discussion to situate our results within a developmental framework, highlighting how early intrinsic connectivity may scaffold alternative trajectories shaped by either visual experience or blindness. The revised text now reads as follows:

“Conversely, for people who remain blind throughout life, visual-PFC connectivity could enable recruitment of visual cortices for higher-order non-visual functions, such as language and executive control (Bedny et al., 2011; Kanjlia et al., 2021). Our results suggest that blind adults may build on connectivity patterns already present in infancy: like blind adults, sighted infants show stronger occipital–PFC than occipital–sensory–motor coupling. Repeated engagement of occipital networks during higher cognitive tasks in early development could intern enhance connectivity and specialization of visual networks for non-visual higher-order functions.

Some prior studies have measured resting-state and task-based functional profiles in the same participants. These studies find that within visual cortices of blind people, the task-based profile of a cortical area is related to its resting state connectivity pattern (citations.) This suggests that these two measures are related. However, the timecourse of this relationship, the developmental trajectory and mechanism of plasticity is not known. Primarily this is because there is very little relevant developmental evidence. For example, in the current study we find that the resting state profile of secondary visual networks in infants is similar to that of blind adults. However, we do not know whether the visual cortices of infants show enhanced task-based cross modal responses, relative to sighted adults and how this compares to responses observed in blind adults. Future work with infants and children would be able to address this question.

In the current study, the clearest evidence for functional change driven by blindness was observed for laterality. Connectivity lateralization in sighted infants resembles that of sighted adults, in both V1 and secondary visual cortices. Relative to both sighted infants and sighted adults, blind adults show more lateralized connectivity patterns between occipital and prefrontal cortices. Previous studies suggest that in people born blind occipital and non-occipital language responses are co-lateralized (Lane et al., 2017; Tian et al., 2023). We speculate that habitual activation of visual cortices by higher-cognitive tasks, such as language, which are themselves highly lateralized, contributes to this biased connectivity pattern of occipital cortex in blindness. Taken together, these results suggest a developmental framework in which intrinsic connectivity present in infancy provides a scaffold that is subsequently shaped and reinforced by experience-dependent recruitment, through either visual experience or the lifelong absence of vision in blindness. Longitudinal work across successive developmental stages will be crucial to test how the alternative trajectories shaped by visual experience versus blindness unfold over development.” (page 14-15)

(6) The split-half reliability analysis is a valuable control. Additional details would clarify what these noise ceilings reflect. Were the rsFC patterns for each ROI calculated only for the ROIs included in the current study or was a broader assessment across the whole brain performed? It also would be helpful to report whether reliability differed for individual ROIs within and between groups. Even if global reliability is matched, selective differences could influence group comparisons. Several infants in the dhcp dataset were scanned twice. Were any second scans included in the current analyses? Comparing first versus second scans directly could strengthen the claim that several weeks of visual experience are insufficient to shift connectivity toward a sighted adult profile.

Thanks to the reviewer’s comments on the reliability of the current study.

In the present study, the noise ceiling was computed from the reliability of the ROI-wise FC profiles used across all analyses. Reliability was estimated using a split-half procedure: each rs-fMRI time series was divided into two equal halves, FC among all ROIs included in the study was computed separately for each half, and the noise ceiling for each ROI was defined as the Pearson correlation between its two FC profiles. Then we averaged these ROI-wise noise ceilings to evaluate group-level reliability, which exceeded 0.70 in all three groups and found no significant difference across groups. This provides an estimate of the upper bound on explainable variance for the exact FC features subjected to statistical testing (Lage-Castellanos et al., 2019). A brief description has been added to the manuscript (page 19, line 518).

Regarding the reviewer’s question about the scope of rsFC features used in the noise-ceiling analysis: we computed noise ceilings only for the ROIs included in the present study, because all analyses in this work were conducted at the ROI–ROI level and did not involve voxelwise whole-brain FC. Thus, the noise-ceiling estimates correspond directly to the full set of FC features on which all statistical comparisons were based.

As suggested by the reviewer, we examined noise ceilings for each ROI separately. All ROIs showed high absolute reliability (noise ceiling > 0.80) across the three groups, indicating that the ROI-wise FC estimates are generally robust across participants. Although many ROIs exhibited statistically significant group differences in noise ceiling (one-way ANOVA, p < 0.05), the effect sizes were small to moderate (partial η^2^ < 0.14). These differences indicate that reliability may vary modestly across groups at the ROI level, and we cannot fully determine whether such variability contributes to the observed different FC patterns across groups. We have included this point in the revised manuscript (page 19, line 525), along with the full statistical results for the ROI-wise noise ceilings in the Supplementary Table S2.

Last, we fully agree that longitudinal comparisons across multiple time points can provide important insights into how early visual experience shapes connectivity. At the same time, in the present dataset, the first scan occurred at a preterm age and the second at term-equivalent age. The differences between the first and second scans would reflect not only additional weeks of visual input, but also differences in prematurity status and overall neurodevelopmental maturity, which would make the interpretation of such comparisons difficult in the context of our current aims. We have clarified in the revised manuscript that only term-equivalent (second) scans were included. We see careful longitudinal work as an important avenue for addressing this question more directly.

(7) The signal dropout assessment in the infant dataset is a valuable quality control step. Applying the same metric to the adult datasets would help harmonize preprocessing across groups and increase confidence in group-level comparisons.

Thank you for this valuable suggestion. Following your comment, we applied the same signal dropout assessment to the adult datasets. One participant in the sighted adult group and two participants in the blind adult group showed signal dropout in one ROI each. The corresponding results are now included in the Supplementary Materials (Figure S13). The findings remain unchanged after this additional control analysis. We also add the relevant content in the Method part as follows:

“The same signal dropout assessment was also applied to the blind and sighted adults to ensure consistent quality control across groups. One participant in the sighted adult group and two participants in the blind adult group exhibited signal dropout in one ROI each. Excluding these participants did not alter the group-level results (see Figure 1–figure supplement 9).” (page 16, line 449)

Minor:(8) The authors added accurate anatomical descriptions to the methods but a less precise characterization remains in the introduction: "Anatomically, these regions correspond roughly to the location of areas such as motion area V5/MT+, the lateral occipital complex (LO), V3a and V4v in sighted people."

We thank the reviewer for this helpful comment. We have revised the Introduction to provide a fuller anatomical description, consistent with the Methods. The text now reads:

“Anatomically, these regions in sighted people approximately correspond to the locations of motion-sensitive V5/MT+ and the lateral occipital complex (LO), as well as ventral portions of occipito-temporal cortex including V4v and dorsal portions including V3a. The occipital ROI also extends ventrally into the middle portion of the ventral temporal lobe and dorsally into the intraparietal sulcus and superior parietal lobule.” (page 3, line 88)

(9)Typo: "lager effect" should be "larger effect."Secondary visual cortices showed a significant within > between difference in both groups, with a lager effect in the blind group (post-hoc tests, Bonferroni-corrected paired: t-test: sighted adults within hemisphere > between hemisphere: t (49) = 7.441, p = 0.012; blind adults within hemisphere > between hemisphere: t (29) = 10.735, p < 0.001; V1: F(1, 78) = 87.211, p < 0.001).

We thank the reviewer for catching this typo. We have corrected “lager effect” to “larger effect” in the revised manuscript. (page 9, line 214)

**Reviewer #2 (Recommendations for the authors):**
All of my other concerns were adequately addressed.

We thank the reviewer for their positive evaluation, and we are glad that our revisions have addressed their concerns.

**Reviewer #3 (Recommendations for the authors):**
In my view, qualifying infants as "sighted" is confusing and unnecessary: why not simplifying and homogenizing the wording along the manuscript and figures?

We thank the reviewer for this suggestion. We agree and have revised the manuscript to use consistent wording, avoiding the qualification of infants as “sighted.”

l188, I don't understand the sentence "By contrast, in sighted adults, this cross-hemisphere difference is weak or absent."

We thank the reviewer for noting that this sentence was unclear. We have revised the text to provide a more precise explanation. The text now reads:

“By contrast, in sighted adults this lateralized pattern is weaker: visual areas in each hemisphere show only a modest preference for ipsilateral prefrontal cortices, and connectivity with the contralateral PFC remains comparatively strong.” (page 8, line 207)

l193: "Secondary visual cortices showed a significant within > between difference in both groups, with a lager effect in the blind group": providing effect sizes for the 2 groups would strengthen this result (+ note the typo laRger).- Figure S7, S11: Please add titles of y-axes.

Thank you for this helpful suggestion. We have corrected the typo and added the effect sizes for both groups in the revised text. The revised sentence now reads as follows:

“Secondary visual cortices showed a significant within > between difference in both groups, with a larger effect in the blind group (post-hoc tests, Bonferroni-corrected paired: t-test: sighted adults within hemisphere > between hemisphere: t (49) = 7.441, p = 0.012, cohen’d = 0.817; blind adults within hemisphere > between hemisphere: t (29) = 10.735, p < 0.001, cohen’d = 1.96).” (page 9, line 214)

Titles of the y-axes have also been added to Figure 2–figure supplement 2 and Figure 1–figure supplement 7.